# Revealing the diversity of commensal corynebacteria from a single human skin site

Reyme Herman,[1,2] Sean Meaden,[1] Michelle Rudden,[3] Robert Cornmell,[4] Holly N. Wilkinson,[3] Matthew J. Hardman,[3] Anthony J. Wilkinson,[5] Barry Murphy,[4] Gavin H. Thomas[1,2]

**ABSTRACT** Our understanding of the skin microbiome has significantly advanced with the rise of sequencing technologies. While *Corynebacterium* species are a major component of the human skin microbiome, research has largely centered on other prevalent genera like *Staphylococcus* and *Cutibacterium*. Prior to this study, complete genomes for skin-associated *Corynebacterium* were limited. For instance, only nine were available for the commonly identified species *Corynebacterium tuberculostearicum*. In this study, we explored *Corynebacterium* diversity from a single skin site, the axilla, using selective media to enrich for these bacteria. Long-read whole-genome sequencing and bioinformatic analysis of the enriched isolates provided unprecedented insight into the diversity of this genus at a single body site. The study yielded 215 closed genomes, comprising 30 distinct representative genomes following dereplication. These genomes span seven distinct species, including two new species provisionally named *Corynebacterium axilliensis* and *Corynebacterium jamesii*, as well as species not previously linked to the skin. Pangenome analysis of these isolates uncovered potential metabolic differences, antimicrobial resistance genes, novel biosynthetic gene clusters, prophages, and phage defense systems. This study represents the most detailed analysis to date of *Corynebacterium* from a single skin niche and highlights extensive variation even within a single host. Our culture-based Nanopore sequencing approach has expanded the number of publicly available skin *Corynebacterium* genomes, providing a valuable resource for future studies investigating the diversity and function of this important skin genus.

**IMPORTANCE** This study uncovers the hidden diversity of *Corynebacterium*, an important yet often overlooked group of bacteria inhabiting human skin. Focusing on the underarm and using advanced sequencing techniques, we identified over 200 complete bacterial genomes. This collection includes species not previously known to reside on the skin, as well as two entirely new species, highlighting how little is currently known about this cutaneous genus. Most of these bacteria were isolated from a single individual, underscoring the vast microbial diversity that can exist within one person. By closely examining the corynebacterial community at a single site, we begin to uncover the complex relationships within a true microbial ecosystem. These findings deepen our understanding of the skin microbiome and provide a valuable resource for future research into how these microbes affect skin health, hygiene, and disease.

**KEYWORDS** skin microbiome, *Corynebacterium*, whole genome sequencing, comparative genomics

**Peer Reviewers** Holger Brüggemann, Aarhus University, Aarhus, Denmark; Irma Martinez-Flores, Centro de Ciencias Genomicas, Cuernavaca, Morelos, Mexico

Address correspondence to Gavin H. Thomas, gavin.thomas@york.ac.uk.

The authors declare that B. M. and R. C. are current employees of Unilever.

See the funding table on p. 15.

Exploration of the human skin microbiome allows deeper understanding of how commensal microbes and the host interact to maintain skin health. Human skin, generally a nutrient-limited environment, hosts a variety of commensal microbes

spanning multiple genera. Skin sites can be characterized as dry, moist, or oily (sebaceous), depending on the architecture of the skin and abundance of associated glands, namely eccrine, sebaceous, or apocrine glands. These different environmental conditions are known to play a major role in determining microbial diversity (1). Although *Cutibacterium* and *Staphylococcus* are widely reported as major commensal genera, *Corynebacterium* are also highly abundant (2, 3). For example, Grice et al. report that corynebacteria comprise 22.8% of 112,000 16S rRNA sequences derived from 20 different skin sites, second only to cutibacteria (23%) (4). Some corynebacterial species have been found to be lipophilic and hence require lipids derived from the sebaceous gland or the stratum corneum (5, 6). The commonly identified commensal corynebacteria *C. jeikeium* (7), *C. amycolatum* (8), and *C. kroppenstedtii* (9) have been suggested to be opportunistic pathogens. *C. accolens* (10) and *C. tuberculostearicum* (11) have been shown to interact with the human immune system, causing inflammation. *C. accolens* has also been shown to inhibit *Streptococcus pneumoniae* (12, 13) while *C. striatum* is proposed to be able to shift *Staphylococcus aureus* from a virulent to commensal state (14), thus demonstrating its potential protective capabilities. Together, these studies highlight the importance of the skin microbiome in protecting against pathogens and maintaining skin homeostasis.

Human malodor is generated by microbial metabolism of a precursor secreted by the apocrine glands. Specific skin-associated bacteria, such as *Staphylococcus* species, convert these odorless precursors into volatile compounds responsible for body odor. Although a monophyletic group of staphylococci was attributed to this phenomenon (15), some studies have proposed a role for axillary *Corynebacterium* in human malodor formation due to the correlation of abundance and odor profiles (16) and the presence of enzymatic components for the biotransformation of the human-derived precursor (17–19). Furthermore, a genus-level study analyzing the diversity of axillary commensals in nine individuals found that 59.5% of sequences were binned to the genus *Corynebacterium* (20). However, our understanding of the role of cutaneous corynebacteria in human malodor remains unclear.

Despite the abundance of corynebacteria on the human skin, the amount of genetic information on cutaneous corynebacterial species is surprisingly lacking, suggesting an incomplete understanding of true diversity within the skin microbiome. In the case of *C. tuberculostearicum*, a species widely found on various skin sites, only nine complete genomes were publicly available on GenBank prior to this study. In comparison, there are 306 available complete genomes of another skin commensal, *Staphylococcus epidermidis* (data from 28 March 2025). Two recent comparative genomics studies have sought to address the lack of genetic information through a culture-based approach. The first focuses on skin isolates of the *C. tuberculostearicum* species complex, which is made up of multiple closely related species including *Corynebacterium kefirresidentii* (21). In addition, the study reports the identification of a novel species of corynebacteria that was predominantly found on the toe web and toenails of multiple individuals. A second comparative genomics study on human skin corynebacteria (22) revealed the diversity of corynebacteria on human facial skin. This study also identified two novel species of corynebacteria.

Here, we sought to elucidate the diversity of corynebacteria within the human underarm (axilla). This is a moist skin site enriched with metabolites derived from the three skin-associated glands as well as the epidermis, making it a relatively nutrient-rich environment compared to most other skin sites. In this study, we used a culture-based approach followed by sequencing, consistent with recent comparative genomics studies. However, those studies typically began with 16S rRNA (V1-V3 region) sequencing to identify representative taxa within amplicon sequence variants for subsequent whole-genome sequencing (WGS). We instead adopted a direct-to-long-read WGS strategy by omitting the 16S rRNA gene sequencing, which could introduce sampling bias. In this study, we generated complete genomes of over 200 corynebacteria isolates derived from axillary swabs of four individuals using the Oxford Nanopore PromethION platform. Our

sequencing approach revealed seven distinct species of axillary corynebacteria, two of which are proposed to be novel species. Through comprehensive genomic analyses, we revealed the genetic and functional diversity within our collection of axillary corynebacteria, paving the way to revealing their roles in the skin microbiome.

## RESULTS

### High-resolution genomic characterization reveals corynebacterial diversity in the human axilla

Axillary swabs from four human volunteers (A1, A3, A4, and A5) were collected and then enriched for corynebacteria on selective agar. To assess the ability of our approach to capture the diversity of corynebacteria within our isolates, we generated complete genome and plasmid sequences for 154 corynebacteria isolates from a single individual, A1, and complemented the study with small subsets of isolates from the three other volunteers: 22 isolates from A3, 26 isolates from A4, and 13 isolates from A5 (Fig. 1A).

In order to initially assess the diversity of the corynebacterial isolates from all four volunteers, we compared pairwise average nucleotide identities (ANI) between assembled genomes from each individual (Fig. 1B through E) and collectively (Fig. S1). The pairwise percentage identity plots for the 154 isolates from A1 revealed that the majority of isolates recovered by culture belonged to two discrete groups (labeled as I and II) (Fig. 1B). However, the depth of sampling also revealed a third cluster (group III) and a number of other smaller groups, some only containing one isolate, suggesting the approach had recovered significant diversity from this single body site from a single individual. For individuals A3 and A5, there was a similar pattern of a number of different clusters identified, despite there being a much smaller total number of genomes (22 and 13 isolates, respectively) (Fig. 1C and E). Surprisingly, in individual A4, we found that almost all of the 26 isolates were very similar to each other (ANI = >95%), suggesting that one species dominates this community (Fig. 1D).

Using the drep workflow (23), we identified representative genomes from groups of genetically similar isolates. Genome-wide ANI values of <95% initially classify isolates into different species, while further refinement using ANI values of 95%–99.5% distinguishes between different strains (Fig. S2). The chosen ANI thresholds are consistent with recent studies on the diversity of cutaneous corynebacteria (21, 22). The isolates were binned to seven different species with 30 identified strains, which we focused further analysis on. Typically, 16S rRNA (V1-V3 region) sequences are analyzed for species-level identification of isolates. Using this method, we observe only five distinct clades instead of seven (Fig. S3A). Furthermore, the 16S rRNA (V1-V3 region) sequences of the 24 isolates forming the largest clade were found to be ≥99.8% identical to three closely related but ultimately different species of corynebacteria (Fig. S3A, red shading). This highlights the limitations of 16S rRNA analysis in accurately resolving species-level differences. Hereafter, the representative genomes will be referred to with the prefix YSMA (York Skin Microbiome: Study A).

While the 16S information alone could not accurately speciate our isolates, we were able to achieve this with the closed genomes using GTDB-tk (24, 25) (Fig. S3B; Table S1). This analysis further classified the 24 isolates into three separate species: *C. kefirresidentii, C. tuberculostearicum,* and *Corynebacterium aurimucosum*_E. These three species are commonly found on the skin and are thought to be part of the *C. tuberculostearicum* species complex, which was previously reported (21, 26), confirming the broader significance of these organisms across multiple human skin sites. ANI analysis of the GTDB representative for the species *C. aurimucosum*_E suggests that the species is misassigned and matches more closely with *Corynebacterium marquesiae* (27). From this point onward, we classify our isolates that matched with *C. aurimucosum*_E on GTDB as *C. marquesiae.* Although two isolates (YSMAA1_1_B1 and YSMAA5_1_F11) matched closest with *C. tuberculostearicum* and *C. tuberculostearicum*_C, they were not classified to a specific species due to the predefined ANI reference radius of the analysis. As

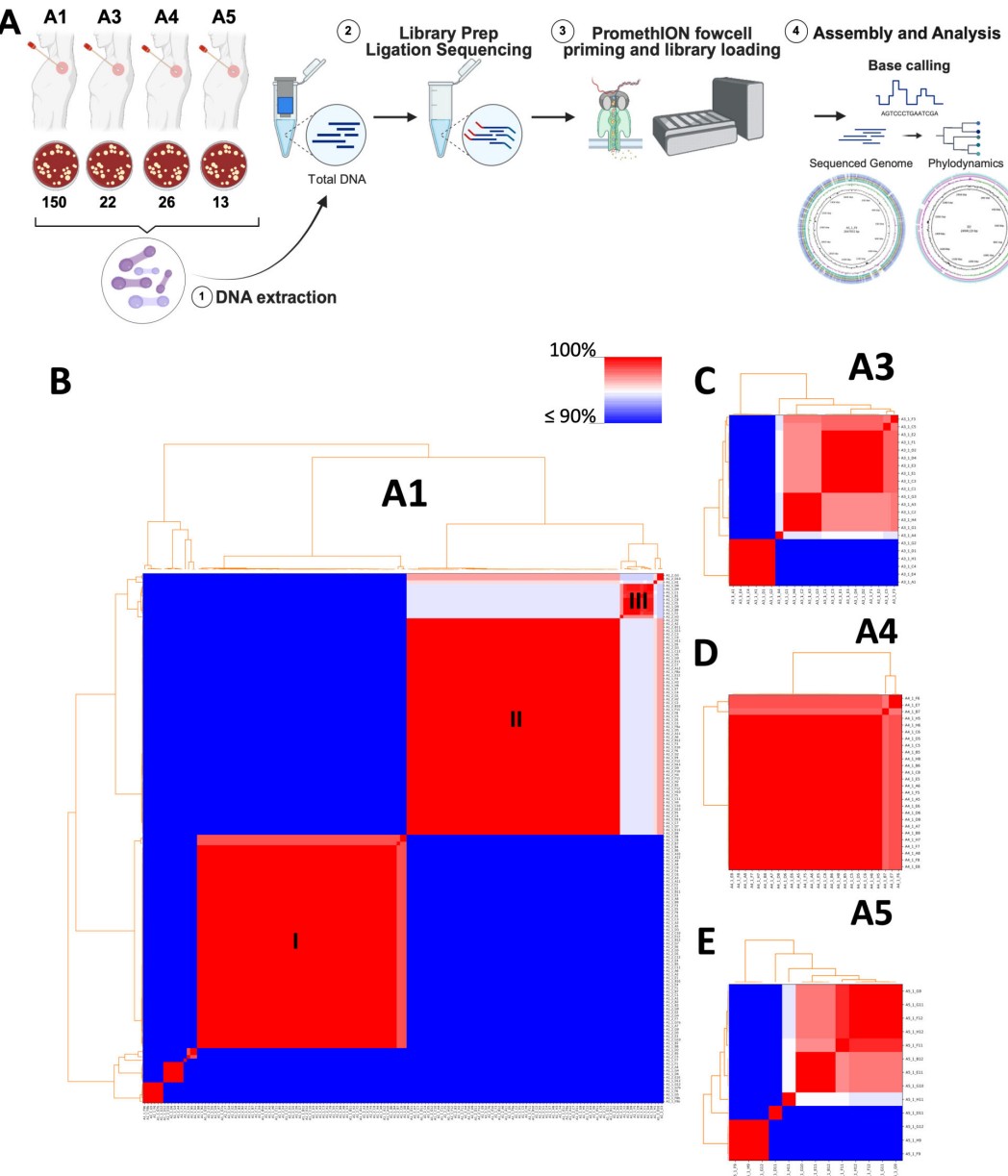

**FIG 1** Capturing the diversity of axillary corynebacteria. (A) Axillary *Corynebacterium* isolates were harvested from swabs from four volunteers. Long-read WGS and downstream analysis was performed on 154 isolates of a volunteer A1, 22 from volunteer A3, 26 from volunteer A4, and 13 from volunteer A5. (B–E) Pairwise average nucleotide identities (ANI) were calculated with pyANI and used to assess the genetic diversity of the isolates. ANI values were placed onto a colored pairwise matrix according to the color scale depicted in (B) for isolates derived from each volunteer. Three major groups of isolates were identified with I, II, and III from the isolates from A1. Images in (A) were created with Biorender.com.

they share more than 95% identity to their closest matches, we classified both as *C. tuberculostearicum*.

YSMAA1_2_B5 and YSMAA1_2_C5 were classified as *C. amycolatum,* which have been identified not only on the skin but also specifically in the human axilla (28, 29) (Table S1). Interestingly, YSMAA1_1_F6 was identified to be *Corynebacterium gottingense,* which was first reported in 2017 (30) but until now has not been associated with the human skin. Finally, three isolates (YSMAA1_1_D6, YSMAA5_1_F9, and YSMAA1_1_F7) were matched with metagenome-derived genomes of uncultured isolates, suggesting that these are two new species. Isolates YSMAA1_1_D6 and YSMAA5_1_F9, tentatively named *Corynebacterium axilliensis*, are most closely related to *Corynebacterium singulare* with an

average nucleotide identity of ~84% calculated using FastANI (31) (Fig. S4A). *C. axilliensis* isolates were found on individuals A1 (~4% of corynebacteria genomes) and A5 (~23% of corynebacteria genomes). Isolate YSMAA1_1_F7, tentatively named *C. jamesii*, is most similar to *C. jeikeium* with an average nucleotide identity of ~87.5% calculated using FastANI (31) (Fig. S4B).

From the four volunteers, our culturomics approach identified 30 genetically distinct isolates spanning seven species of *Corynebacterium* from an initial pool of 215 isolates (Tables S2 to S5). Using our direct to WGS workflow, we revealed the species diversity of our axillary isolates from the four volunteers, which would not be captured if a prior 16S rRNA screen was carried out (*C. tuberculostearicum*, *C. kefirresidentii,* and *C. marquesiae*). Alongside other known cutaneous species (including *C. amycolatum*), we were also able to identify species not previously associated with healthy human skin (*C. gottingense*), including two novel species (*C. axilliensis* and *C. jamesi*).

## Pangenome and metabolic profiling reveal species- and strain-level functional diversity among cutaneous corynebacteria

We performed pangenome analysis of the 30 representative isolates to understand their genetic differences. Firstly, we used Roary (32) to derive a core genome maximum likelihood single nucleotide polymorphism (SNP) tree using a BLASTp cutoff of 80% and seeded the tree with species representatives from GenBank (Fig. S5). Consistent with the GTDB-tk analysis, we clearly see seven different species, including distinctions between the isolates from the species of the *C. tuberculostearicum* species complex and obvious divergence between the two novel species and their nearest related species. To further elucidate the genetic differences between our isolates and their closest GenBank representatives, we performed a pangenome analysis using anvi'o (33) (Fig. 2). Gene clusters of interest were then functionally predicted with eggNOG-mapper (34, 35).

The two representative *C. axilliensis* isolates share a specific cluster of genes with the largest proportion of genes with assigned functions found to be associated with transcription (K) or replication, recombination, and repair (L) (Fig. S6A). Interestingly, analysis of the gene clusters specific to either *C. axilliensis* YSMAA5_1_F9 or YSMAA1_1_D6 places the largest proportion of genes with annotated functions in the cell wall/membrane/envelope biogenesis (M) category, suggesting differences in their cell envelope (Fig. S6B and C). Functionally annotated gene clusters specific to *C. jamesii* YSMAA1_1_F7 are mainly found in the replication, recombination, and repair (L) (Fig. S7).

As expected, *C. kefirresidentii*, *C. marquesiae,* and *C. tuberculostearicum* isolates share a larger number of gene clusters than with other species. To clarify the main differences between the three species, we generated the core genomes for each species from isolates within our collection followed by a pangenome analysis (Fig. S8) to extract the singletons found within each species for functional analysis. Within the functionally predicted singletons, genes classified in the inorganic ion transport and metabolism (P) category make up the largest proportions in each of the three core genomes (Fig. S9A through C), consistent with a previous study on skin isolates of this species complex (21).

We further investigated the putative metabolic differences between these isolates by identifying complete metabolic pathways present in each skin isolate using the kyoto encyclopedia of genes and genomes (KEGG) database (Fig. 3). As observed in the pangenome analysis, we generally observe KEGG modules clustering with more similar species. Within the *C. tuberculostearicum* species complex isolates, we see conservation of most KEGG modules, but the largest differences seem to lie within the amino acid metabolism pathways. Unlike other species within the complex, most *C. maquesiae* isolates do not seem to possess the serine biosynthetic module (M00020), while *C. tuberculostearicum* lacks both lysine biosynthetic modules (M00016, M000526) and the ornithine biosynthesis (M00028) modules, and *C. kefirresidentii* lacks the tryptophan biosynthetic module (M00023).

The tryptophan biosynthetic module (M00023) seems to also be absent in all isolates of *C. gottingense*, *C. jamesii*, *C. amycolatum,* and *C. axilliensis* YSMAA1_1_D6. Interestingly,

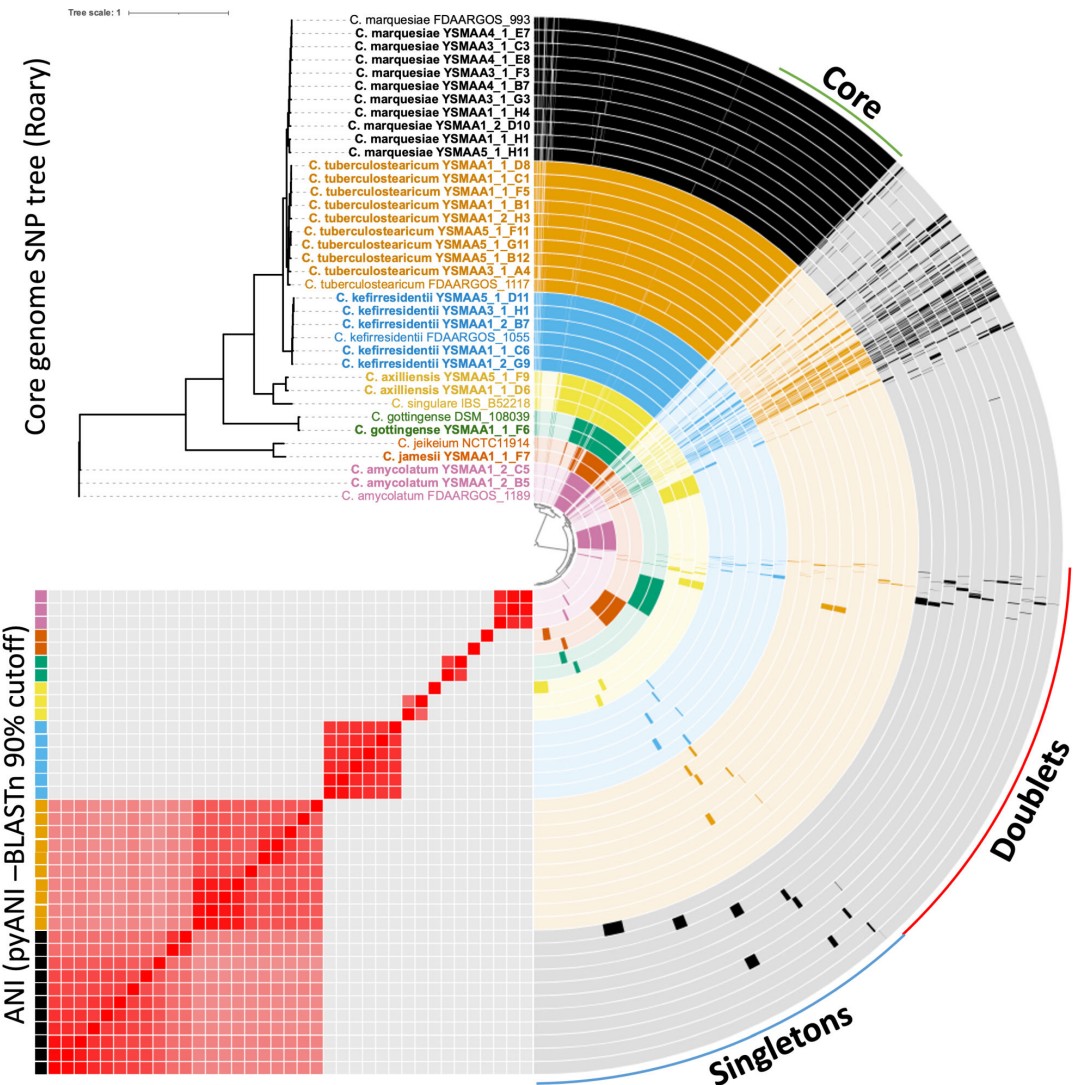

**FIG 2** The pangenome of the axillary corynebacteria isolates (bold) with the closest RefSeq representatives for each species calculated and visualized using anvi'o. Gene clusters found in either one (singletons) or two (doublets) were identified. A core genome was calculated using Roary with an 80% BLASTp cutoff and visualized on a tree generated using PhyML and visualized on iTOL. An ANI comparison using BLASTn on pyANI with a 90% cutoff was generated and visualized on anvi'o. Isolates of the seven clusters of species were color-coded (*C. tuberculostearicum* = orange, *C. kefirresidentii* = blue, *C. axilliensis* = gold, *C. gottingense* = green, *C. jamesii* = bronze, *C. marquesiae* = black, *C. amycolatum* = pink).

the two isolates of *C. axilliensis* seem to possess different metabolic pathway profiles, not limited to tryptophan biosynthesis. Another example is the fatty acid biosynthesis module (M00083), found only in *C. axilliensis* YSMAA5_1_F9 and both *C. amycolatum* isolates.

Another category of metabolic pathways in which we observe a large number of differences is the metabolism of cofactors and vitamins. All but two isolates of *C. kefirresidentii* seem to be biotin auxotrophs, while *C. gottingense* and *C. amycolatum* isolates were suggested to be pyridoxal phosphate auxotrophs. Finally, *C. amycolatum* YSMAA1_2_B5 seems to be the only isolate that is able to synthesize pantothenate from valine or aspartate (M00119).

Overall, these differences, including intra-species variation, are captured in the 16 corynebacteria isolates from a single individual, A1 (Fig. S10). Although we have observed differences between isolates of the same species, we identified mainly

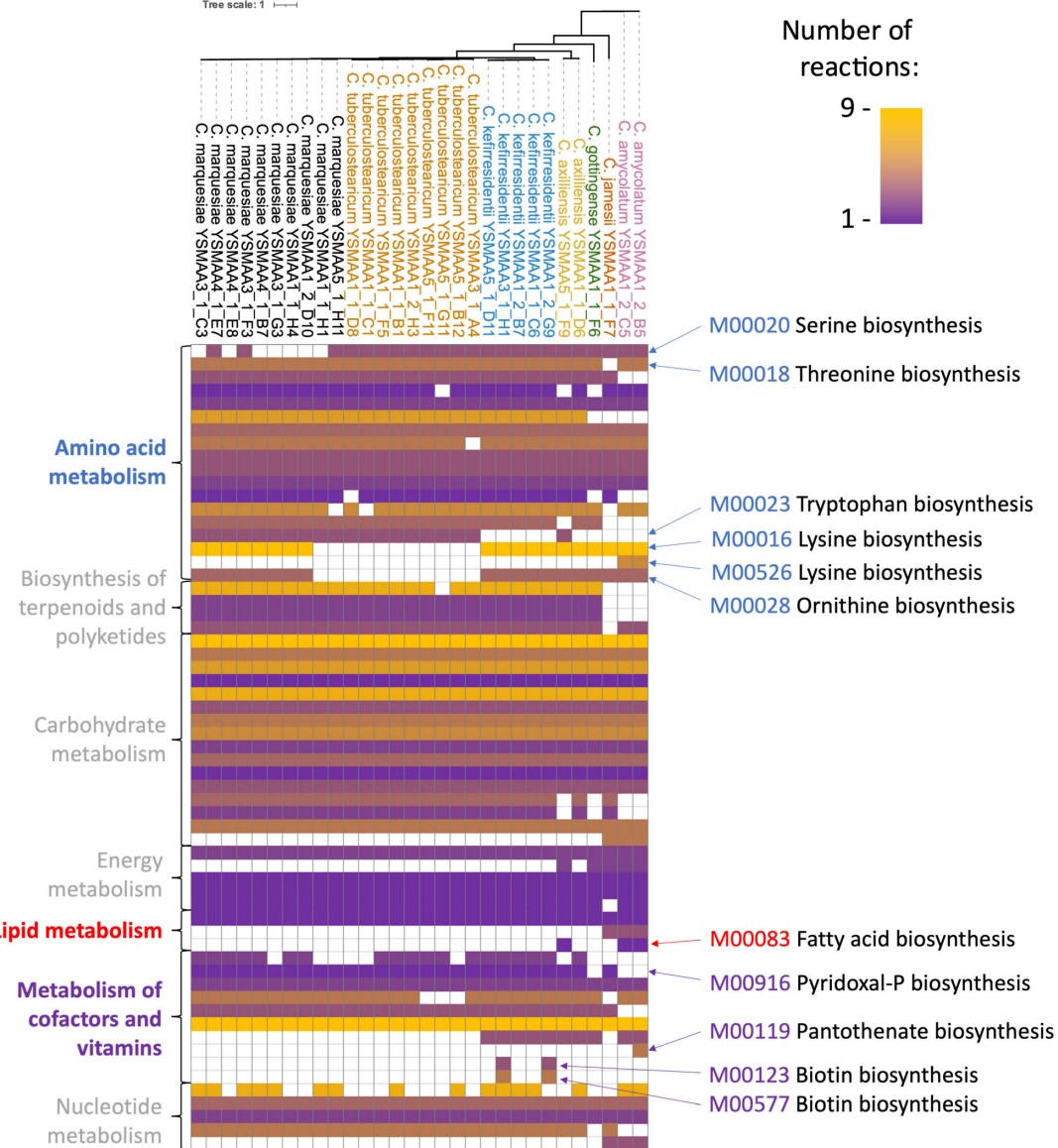

**FIG 3** Metabolic pathways in axillary corynebacteria reveal species-specific general profiles and some variation within species. Each KEGG module for each isolate was placed on a heatmap, where the modules with the largest number of reactions are in yellow and lowest in purple, with absent modules in white. Modules of interest are highlighted on the right of the heatmap. KEGG modules were grouped according to the categories on the left of the heatmap. Isolates were grouped by species with a core genome tree depicting the variation. Species names were colored accordingly (*C. tuberculostearicum*=orange, *C. kefirresidentii*=blue, *C. axilliensis*=gold, *C. gottingense*=green, *C. jamesii*=bronze, *C. marquesiae*=black, *C. amycolatum*=pink).

species-specific metabolic pathways that could be used to understand the potential to modulate the cutaneous corynebacterial community through metabolic means.

## Diverse biosynthetic capacities and antimicrobial resistance in corynebacterial skin isolates

Bacteria often produce complex small molecules that have various functions relating to microbial competition such as antibiotics and siderophores, which are encoded within recognizable biosynthetic gene clusters (BGCs). Resident bacteria of highly complex microbial communities like the skin are known to produce antibiotics to suppress other species in their niche, including preventing colonization by pathogenic species (36) and, conversely, allowing for the proliferation of others (37). We used antiSMASH to

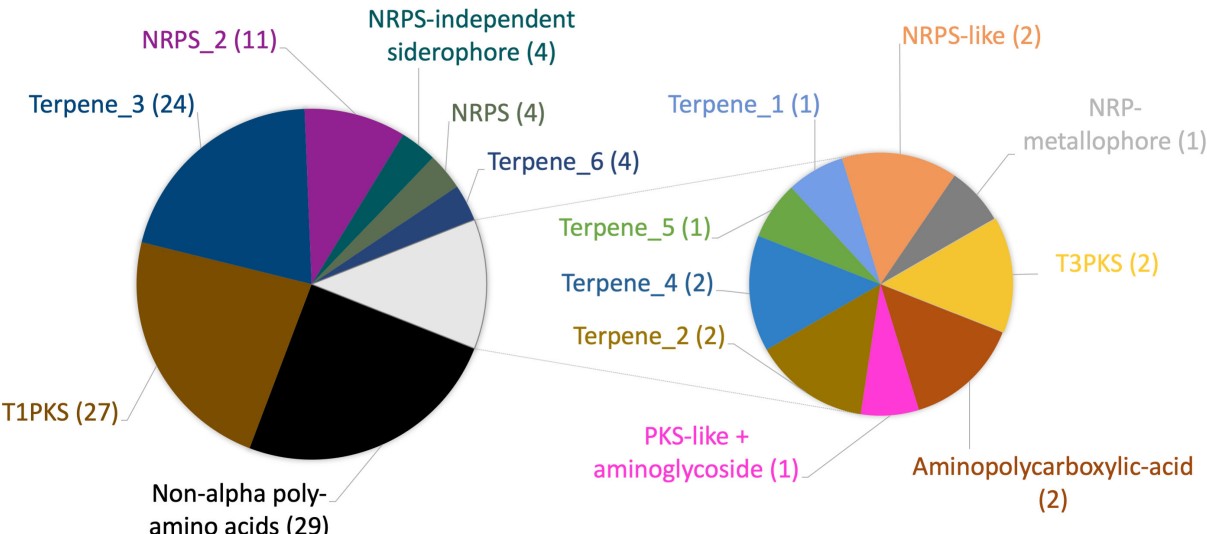

**FIG 4** Multiple putative BGC types were identified in the representative isolate genomes using antiSMASH 7. Names of the different BGC types were included with different cluster architectures in numbered parentheses (i.e., [Name] (X), where X corresponds to a cluster architecture type). The prevalence of each BGC type in the 30 dereplicated genomes was also included after the names of each BGC in brackets.

identify putative BGCs within the representative set of isolates (38). For the purposes of this global corynebacteria BGC analysis, we group similar predicted clusters together and determine the prevalence of each discrete cluster (Fig. 4). The majority of the isolates share a predicted non-alpha poly-amino acid (NAPAA), a type 1 polyketide synthase (T1PKS), and a terpene biosynthetic cluster. Other predicted BGCs include non-ribosomal polyketide synthase (NRPS) and multiple terpene clusters with different architectures (Fig. S11 and S12). The *C. tuberculostearicum* species complex isolates were generally found to possess similar BGCs with *C. tuberculostearicum* isolates having the largest variation (Fig. S13). As expected, distantly related species like *C. amycolatum*, *C. gottingense,* and *C. jamesii* were predicted to have BGCs that are more distinct or of different classes (Fig. S14). Despite being outside the *C. tuberculostearicum* species complex, the analyzed *C. axilliensis* isolates seemed to also share similar predicted BGCs. With the exception of the cationic peptide ε-poly-L-lysine (NAPAA), we were not able to confidently predict the products of these BGCs, as the most similar known clusters for all of the predicted BGCs shared low percentage similarities with their respective putative clusters. Nevertheless, our analysis suggested different BGCs harbored by the corynebacterial isolates, which could contribute to the modulation of the skin microbiome through the production of metabolites like siderophores (e.g., NI siderophore, aminopolycarboxylic acid, NRPS) and even antimicrobials (e.g., T1PKS, PKS-like + aminoglycoside).

The likely presence of antimicrobials on the human skin may promote the horizontal acquisition of genes encoding antimicrobials or the selection of mutants altering the properties of existing cellular proteins. ResFinder analysis revealed 15 of the 30 isolates contained varying numbers and types of acquired resistance genes but no predicted chromosomal mutations leading to resistance (Table S6). The most common resistance gene identified was *erm(X)*, which is proposed to confer resistance to erythromycin-related antibiotics and is widely found across the genus (39–43). *C. tuberculostearicum* YSMAA5_1_F11 and YSMAA5_1_G11 were also found to have an aminoglycoside resistance gene that could confer resistance to gentamicin and tobramycin. *C. marquesiae* YSMAA1_1_H4, *C. tuberculostearicum* YSMAA3_1_A4, and *C. axilliensis* YSMAA5_1_F9 seem to have a resistance gene to an unknown aminoglycoside initially found in *C. striatum*. Interestingly, *C. axilliensis* YSMAA5_1_F9 has genes conferring resistance to chloramphenicol (*cmx*), streptomycin (*aph(6)-Id, aph(3")-Ib*), and kanamycin (*aph(3")-Ia*). These were found in tandem in the same region of the genome surrounded by multiple

insertion sequence (IS) elements and were also not found in *C. axilliensis* YSMAA1_1_D6, suggesting a recent acquisition. Finally, we identified a sole tetracycline resistance gene in *C. tuberculostearicum* YSMAA5_1_B12, which also resides near IS elements, suggesting another example of horizontally acquired resistance mechanisms. We attempted to experimentally verify what we identified using ResFinder using the disk diffusion assay (Table S7). Using clindamycin, we found that 10 of the 30 isolates were resistant, all of which contain an *erm*(X) gene. However, in addition to these 10 isolates, we note that four additional isolates contained an *erm*(X) gene but were not resistant in our conditions. *C. axilliensis* YSMAA5_1_F9 was the only isolate predicted to be resistant to chloramphenicol (via the *cmx* gene), and indeed was the only isolate observed to be resistant in our conditions. *C. tuberculostearicum* YSMAA5_1_B12 was predicted to be doxycycline resistant (*tet*(Z) gene), but did not display the phenotype in our disk diffusion assay. The presence of certain resistance genes may not confer resistance in lab-based disk diffusion assays, as we may not have captured the most appropriate conditions in which these isolates would be resistant. Interestingly, *C. jamesii* YSMAA1_1_F7 was the only isolate that was resistant to penicillin G in our conditions, even though a β-lactamase was not predicted to be in the genome, suggesting this isolate has some other innate mechanism of resistance.

## Diverse prophage repertoires and conserved phage defense systems in skin-associated *Corynebacterium* isolates

Another type of commonly acquired mobile genetic element is prophages. Phages are commonly found in the microbiome and could play an important role in skin health by regulating the bacterial community. We identified 17 putative prophage sequences in our 30 representative genomes, with an average of 0.6 (± 1.4) prophages per genome (Table 1). Putative prophages were identified using geNomad (44), followed by gene annotation using Pharokka and Phold (45). Only putative prophages containing genes that were functionally annotated with multiple common features of a phage (e.g., tail proteins, head, and packaging) were retained for further analysis. These putative prophages were then searched against the NCBI database using BLASTn to identify

**TABLE 1** The closest phages and bacteria genomes were identified using a BLASTn analysis on the putative prophages identified by geNomad[a]

| Putative prophage name | Putative prophage length | Source species | Closest phage hit (% identity, % query cover) |
|---|---|---|---|
| A1_1_C6_phi1_2 | 52,250 | *C. kefirresidentii* | ctNWy2 (92.03%, 69%) |
| A1_1_D6_phi1 | 42,557 | *C. axilliensis* | ctIBL2 (83.98%, 29%) |
| A1_1_F7_phi1 | 56,384 | *C. jamesii* | ctwl93 (77.01%, <1%) |
| A1_1_H1_phi1 | 49,192 | *C. marquesiae* | ctfR81 (90.01%, 67%) |
| A1_2_B5_phi1 | 38,625 | *C. amycolatum* | – |
| A1_2_D10_phi1 | 42,401 | *C. marquesiae* | ctKpD5 (81.46%, 43%) |
| A1_2_G9_phi1 | 26,374 | *C. kefirresidentii* | ctOSk1 (95.99%, 64%) |
| A3_1_C3_phi1 | 38,823 | *C. marquesiae* | ctSSL1 (93.44%, 45%) |
| A3_1_F3_phi1 | 54,302 | *C. marquesiae* | ctcbl1 (92.73%, 70%) |
| A3_1_G3_phi1 | 47,617 | *C. marquesiae* | ct1BD1 (92.25%, 44%) |
| A3_1_H1_phi1 | 47,347 | *C. kefirresidentii* | ctLCM1 (95.25%, 44%) |
| A4_1_E7_phi1 | 47,738 | *C. marquesiae* | ct1BD1 (86.18%, 54%( |
| A5_1_D11_phi5_2_1 | 46,970 | *C. kefirresidentii* | ctNWy2 (95.17%, 85%) |
| A5_1_D11_phi3_4 | 50,578 | *C. kefirresidentii* | ctNWy2 (96.26%, 64%) |
| A5_1_F9_phi1 | 43,162 | *C. axilliensis* | ctIBL2 (84.39%, 19%) |
| A5_1_H11_phi1 | 50,212 | *C. marquesiae* | ctXFF1 (97.39%, 85%) |
| A5_1_H11_phi2 | 49,906 | *C. marquesiae* | ct1BD1 (92.69%, 46%) |

[a]The percentage sequence identity and percentage coverage of each hit were included. The putative prophage names are colored consistently with the species of origin (*C. kefirresidentii* = blue, *C. axilliensis* = gold, *C. jamesii* = bronze, *C. marquesiae* = black, *C. amycolatum* = pink). "–" indicates no phage hits were detected in the BLASTn analysis.

the closest related phage and bacteria species. Some putative prophages were either matched with phages but retained poor coverage (A1_1_F7_phi1, A5_1_F9_phi1) or were not matched with any (A1_2_B5_phi1). As expected, various regions of these putative prophages also matched with other corynebacteria, and similar to prophage hits, we observed some bacteria hits with poor coverage (A1_1_F7_phi1, A3_1_H1_phi1). As some of these isolates were derived from the same skin site of the same individual, we sought to understand the diversity of these putative prophages, as some prophages may be shared between isolates. We performed a multisequence alignment using MAFFT (46) to determine the genetic identities between these putative prophages (Fig. S15). Neither of these putative prophages seems to be shared between isolates, even within the isolates from the same individuals. Remarkably, none of the nine representative *C. tuberculostearicum* genomes were found to possess any putative prophages.

In addition to the presence of prophages, we analyzed the representative genomes with PADLOC (47) to identify potential phage defense systems (Fig. 5). Types I, II, IIG, and IV restriction-modification (RM) systems are commonly found in all isolate genomes, while the type III RM system is much less abundant. Other commonly identified systems include the plasmid transformation protection system Wadjet Type I (48) and the abortive phage infection system AbiD (49). We also observed the CRISPR-Cas Type I-E system in some isolates of *C. amycolatum*, *C. tuberculostearicum,* and *C. jamesii,* and also the CRISPR-Cas Type I-G system only in *C. gottingense*. We also predicted the presence

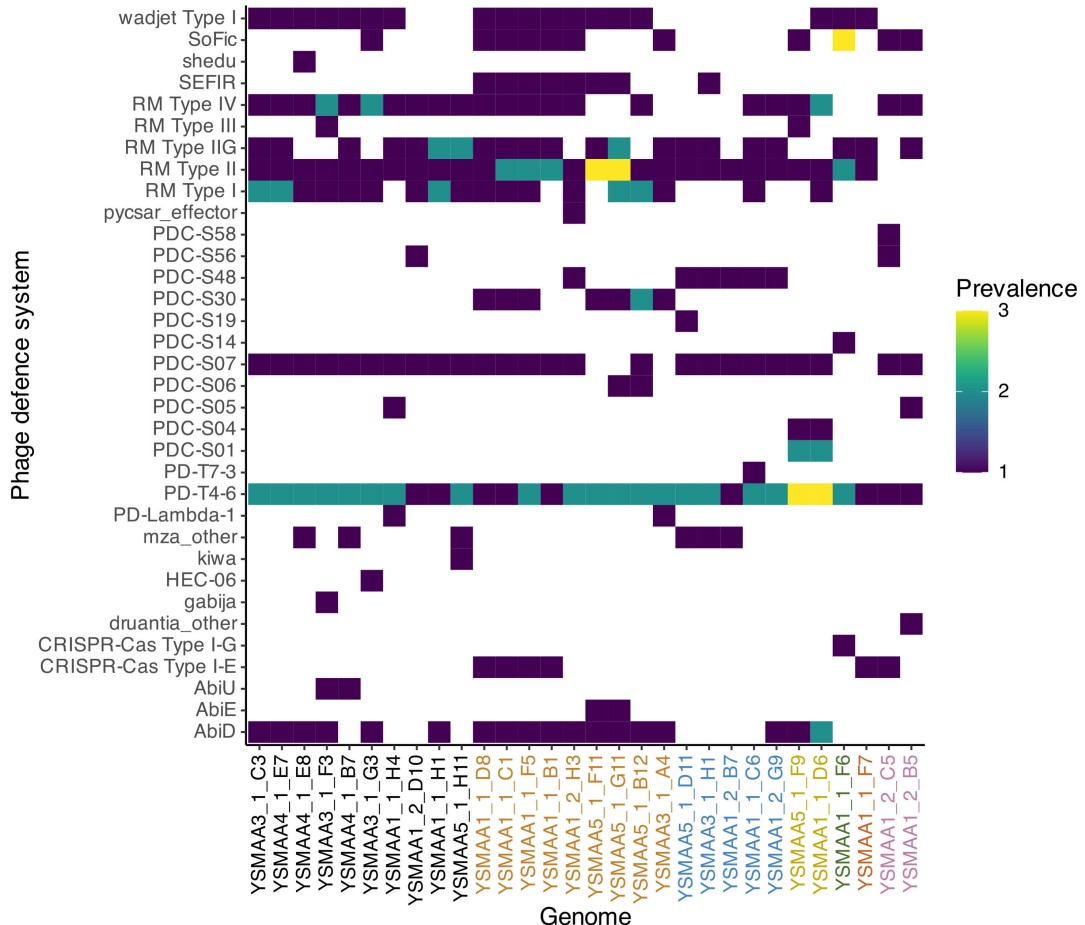

**FIG 5** Various phage defense systems were predicted in the representative genomes using PADLOC. The heatmap depicts the prevalence of each phage defense system within each isolate according to the scale on the right with absent systems in white. The genomes were arranged according to the different corynebacteria species and colored accordingly (*C. tuberculostearicum* = orange, *C. kefirresidentii* = blue, *C. axilliensis* = gold, *C. gottingense* = green, *C. jamesii* = bronze, *C. marquesiae* = black, *C. amycolatum* = pink).

of multiple low-prevalence phage defense systems like the putative nucleotide-sensing Gabija defense (50), the pyrimidine cyclase system for anti-phage resistance (Pycsar) (51), and other defense systems of unknown mechanisms like shedu and kiwa (48). Global PADLOC analysis of all 215 complete genomes of our axillary corynebacteria isolates suggested typical conservation of these predicted phage defense systems within the drep secondary clusters with some exceptions (Fig. S16 to S19). Some of these differences include select isolates from *C. marquesiae* cluster 4_5 and 4_8 and *C. tuberculostearicum* cluster 3_2, which lack the RM type IV system. A minority of *C. marquesiae* cluster 4_5 isolates were also predicted to have the putative phage RNA-interacting Mokosh Type II system (52). Similarly, only two isolates of *C. kefirresidentii* cluster 2_5 lack the RM Type IIG system, and two other isolates of the same cluster were predicted to have the AbiD system. Together, we have comprehensively covered the range of phage defense systems predicted to be present in our axillary isolates, and we expect these mechanisms to be present in other skin-associated corynebacteria.

## DISCUSSION

The growing number of recent studies exploring the diverse roles of skin commensals, whether antagonistic or mutualistic, underscores the need to accurately characterize the true constituents of the skin microbiome. Within the population of skin commensal microbes, staphylococci and cutibacteria have been widely studied (2–4, 53). In contrast, although corynebacteria make up a large proportion of the microbiomes across multiple skin sites, commensal species from this genus remain poorly studied. This highlights the importance of studies like this, not only to uncover the roles of these commensals but also to discover novel skin-associated species.

Others have also sought to address the lack of information on this genus by attempting to clarify the diversity of cutaneous corynebacteria (21, 22, 26), but the genetic information for this genus still lags behind that of cutaneous staphylococci and cutibacteria. The recent advances in long-read sequencing, particularly the R10.4.1 flow cell and Kit 14 chemistry by Oxford Nanopore Technologies, have allowed for high-quality sequencing to be more accessible at lower costs to researchers. With this, we developed and optimized a direct-to-sequencing method, enabling us to contribute another 30 representative cutaneous corynebacteria genomes with another 195 closed genomes that were filtered out during dereplication but are available for analysis.

Unlike our study, previous efforts to understand the skin microbiome usually involved an initial 16S rRNA amplicon sequencing step which serves as a filtering step to select specific isolates for whole-genome sequencing (21, 22). The high-throughput nature of 16S rRNA gene amplicon sequencing makes it an appealing first step in determining the diversity of isolates. A common way to compare the 16S rRNA gene sequences is by aligning their respective V1-V3 fragment, as it has been shown to be able to distinguish between species more reliably (54, 55). However, this approach does not resolve the differences between isolates of closely related but ultimately different species. For example, the skin commensals *C. kefirresidentii* strain FDAARGOS_1055 and *C. tuberculostearicum* strain FDAARGOS_1198 share a 16S rRNA (V1-V3 fragment) sequence identity of 99.8% but only share a ~89% genomic sequence identity when calculated using FastANI. Clearly, the difference between the two species is only obvious when whole-genome sequences are available. The difficulties in resolving the diversity of isolates were also previously reported in multiple other studies as microbial "dark matter," or isolates that cannot easily be speciated using the current tools and information available (2, 56, 57). Furthermore, common mobile elements like prophages, which are known contributors to genetic variation, would be missed without a WGS approach. We were able to keep our WGS cost low, negating the need for an initial 16S rRNA gene filtering approach and allowing us to reveal the multitude of invaluable genetic information from WGS.

Using our method, we were able to confirm the widespread presence of the species *C. tuberculostearicum*, *C. kefirresidentii,* and *C. marquesiae* (formerly *C. aurimucosum*_E) (Fig. S20), which together form the *C. tuberculostearicum* species complex (27). In addition,

we found more rarely *C. amycolatum* strains in the axilla, and unexpectedly, we also identified isolates of *C. gottingense*, a species previously not associated with the skin. This species has been isolated from the blood of patients with bacteremia, the cerebrospinal fluid, and urine (30, 58), suggesting that it possibly originates from the skin. In addition to known species, we also identified two novel species tentatively named *C. jamesii* and *C. axilliensis*. Isolates of *C. axilliensis* were found on individuals A1 and A5 and make up a significant 23% of sequenced corynebacterial isolates from the latter, suggesting this novel species could be a major previously unrecognized component of the axillary microbiome of some individuals.

Although the fundamental roles of cutaneous commensal corynebacteria are still poorly understood, there have been increasing studies demonstrating how bacteria from this genus can contribute to colonization protection, including against other human pathogens. For example, *C. striatum* has been demonstrated to shift *S. aureus* toward commensalism through the suppression of the *agr* quorum-sensing system (14). Additionally, Rosenstein et al. (37) have proposed cooperation between corynebacteria and *Staphylococcus lugdunensis* against *S. aureus* in the nasal cavity. These studies have highlighted the importance of understanding the true diversity of cutaneous corynebacteria species, as we may yet uncover more roles that could be exploited to improve tools and treatments for skin health.

Our isolate characterization approach provided us with the ability to predict differences in metabolic pathways, the presence of biosynthetic gene clusters, mobile genetic elements, and bacterial defense systems. For example, there were general correlations in the presence of complete amino acid pathways between species (e.g., serine biosynthesis), but we also observed intra-species differences, including within the two strains of *C. axilliensis* we identified (Fig. 3). To demonstrate the power of our culturomics approach, we performed an unparalleled in-depth characterization of over 150 axillary isolates from the same volunteer, which suggested the possibility of metabolic sharing (e.g., lysine) between predicted auxotrophs and other members of the microbiome (Fig. S10). Furthermore, the discovery of novel biosynthetic gene clusters, including antimicrobials and secondary metabolites, could pave the way toward understanding how certain members of the genus *Corynebacterium* could modulate the microbiome to maintain skin health. Together, these findings could form the basis of further studies to explore networks between members of the same human skin site.

We acknowledge that the findings of this study are limited by the number of volunteers tested and the heavy characterization of only a single volunteer. Furthermore, only one selective media was used at the point of isolation, which could exclude certain isolates of this genus. Nevertheless, our study not only significantly increased the number of *Corynebacterium* genomes available to help reveal their roles in the skin microbiome but also suggests a workflow that can be adapted further in larger-scale studies of this genus. The fact that even in this study we identified two potential new species suggests that there is still more diversity in skin corynebacteria to be discovered, and as a genus, they lag behind other genera like *Staphylococcus* and *Cutibacterium*. Our cost-effective, direct-to-WGS approach on isolates collected from a single skin site seems an efficient strategy, which can be scaled and expanded. The genomes are publicly available through the YSM isolate collection held on the multi-omics research factory (MORF) platform (https://morf-db.org/projects/Public/MORF000083) and represent a rich genomic and biological resource for the wider research community to deepen our understanding of this historically underexplored genus and their role in the healthy skin microbiome.

## MATERIALS AND METHODS

### Sampling procedures

Four adult male volunteers (age range: 32 to 78) were recruited for axillary swabbing at Unilever R&D, Bebington, UK. The volunteers were in good general health, did not use any body products (e.g., deodorants) for 24 hours before the study, were not taking antibiotics, had no active skin conditions on any part of their bodies, had not suffered from eczema in the last 5 years, nor had they ever had psoriasis. Prior to swabbing, the volunteers were "sniff tested" by a panel of testers and scored according to odor levels (high malodor: A1 and A3, low malodor: A4 and A5). Axillary swabs were then collected by swabbing in a linear motion 20 times using eSwabs (Copan) on both underarms and then placed into Amies transport medium.

### Bacterial culturing

Swabs were plated onto aerobic coryneform plates (ACP) solid media (39.5 g $L^{-1}$ blood Agar base no. 2, 3 g $L^{-1}$ yeast extract, 2 g $L^{-1}$ glucose, 5 mL $L^{-1}$ Tween 80, 50 mL $L^{-1}$ defibrinated horse blood, 100 mg $L^{-1}$ fosfomycin) to enrich for corynebacteria and grown for 2 days at 37°C. Alongside the 215 *Corynebacterium* isolates, we also isolated 42 *Staphylococcus* isolates from the species *Staphylococcus capitis* and *Staphylococcus warneri*, which will be reported in a separate study. Colonies were picked and grown overnight in brain-heart infusion broth + 1% Tween-80 (BHIT) at 37°C, shaking at 200 rpm. Cultures were stocked in BHIT supplemented with 10% glycerol in 2 mL 96-well plates.

### DNA extraction

For genomic and plasmid DNA extraction, 0.5 mL of BHIT was inoculated with each isolate in 2 mL 96-well plates and grown for 20 hours at 37°C, shaking at 200 rpm. Cultures were then pelleted by centrifugation before resuspension in 100 μL enzymatic lysis buffer (50 mM Tris-HCl, pH 8, 50 mM EDTA, pH 8, 0.5% Tween 20, 0.5% Triton-X100, 25 mg/mL lysozyme). The mixtures were incubated at 37°C overnight with gentle shaking (<50 rpm). A total of 500 μL of lysis buffer (0.5 M NaCl, 100 mM Tris-HCl, pH 8, 50 mM EDTA, pH 8, 1.5% SDS) was then mixed into each well and incubated at 80°C for 30 minutes in a thermomixer. The mixture was allowed to cool before the addition of 4 μL of 4 mg/mL RNase A Solution (Promega) before incubation at 37°C for a further hour. Cellular proteins were then degraded with 20 μL of 20 mg/mL Proteinase K Solution (Promega) and incubation at 56°C for 15 minutes. Remaining proteins were precipitated with 200 μL of protein precipitation buffer (5 M potassium acetate). The lysates were then clarified by centrifugation at 16,000 × *g* for 5 minutes. DNA was precipitated from the supernatant using 600 μL isopropanol and incubated overnight at 4°C. The precipitated DNA was pelleted by centrifugation at 16,000 × *g* for 5 minutes. The DNA pellets were washed with 600 μL 70% ethanol and then air-dried for 15 minutes. Finally, the DNA pellets were rehydrated with 50 μL nuclease-free water and incubated overnight at room temperature. The DNA concentrations and QC measurements were performed using Qubit (Thermo Fisher) and Nanodrop (Thermo Fisher).

### DNA library preparation, whole-genome sequencing, and assembly

The Oxford Nanopore Native Barcoding Kit 96 V14 was used to prepare and barcode the isolated genomic and plasmid DNA. The recommended protocol was followed with the following changes: (i) half the recommended volume of the DNA repair and end-prep reaction was prepared; (ii) half the recommended volume of the native barcode ligation reaction was prepared. The prepared library was loaded into a PromethION R10.4.1 flow cell, and sequencing was allowed to take place over 72 hours and base-called using the singleplex high-accuracy model, 400 bps on MinKNOW v.23.07.5. The reads were assembled with the EPI2ME Labs wf-bacterial-genomes isolate workflow (Flye

v.2.92-b1786 [59], Medaka v.1.9.1 (Oxford Nanopore Technologies), Prokka v.1.14.5 [60], ResFinder v.4.3.2 [61, 62]). Genome sequences are available through the York Skin Microbiome project in MORF (https://morf-db.org/projects/Public/MORF000083).

## Bioinformatics tools

Pairwise genome comparisons of all 215 assembled genomes were performed using the ANIm (63) analysis on pyANI v.0.2.12 (64). The genomes were dereplicated with drep v.3.4.2 (23) using Mash v.2.3 (65) and MUMmer v.3.0 (66), employing a primary clustering ANI cutoff of 95% to distinguish between different species and a secondary clustering ANI cutoff of 99.5% to capture strain diversity to derive a representative set of genomes. The core genomes were determined using Roary v.3.13.0 (32) with an 80% BLASTp cutoff. PhyML v.3.3 (67) trees 16S rRNA V1-V3 and core genome alignments of the representative genomes were constructed using the HKY85 substitution model with approximate likelihood-ratio tests (aLRT) (68). Trees were visualized on iTOL v.6.9 (69). Speciation using GTDB-tk v.2.3.2 (24) was performed on KBase (25). The genome alignments of *C. axilliensis* and *C. jamesii* were prepared using BLASTn and visualized using BRIG v.0.95 (70). *C. axilliensis* was named after the axilla, the human body site where this species was first isolated. *C. jamesii* was named after Dr. Alexander Gordon James, who is a pioneer in axillary corynebacterial research.

Pangenome analysis was performed using the anvi-pangenome package and visualized on anvi'o v.8 (33). The representative genomes were also compared using pyANI and visualized with the pangenome on anvi'o. Relevant gene clusters were extracted and functionally analyzed with eggNOG-mapper v.2 (34, 35) to determine the respective cluster of orthologous genes (COG) categories. Metabolic pathways were identified using BlastKOALA v.3.0 and KEGG Reconstruct (71, 72). Biosynthetic gene cluster analysis was performed using the default settings on antiSMASH v.7 (38). Putative prophages were predicted using geNomad v.1.8 (44) and then annotated using Pharokka v.1.7.2 and Phold v.0.1.4 (45). Phage defense systems were predicted using PADLOC v.4.3 (73).

## Antibiotic susceptibility testing

Antibiotic susceptibility testing was performed using plate-based disk diffusion assays using the recommended manufacturer's protocol and the European Committee on Antimicrobial Susceptibility Testing (EUCAST) instructions (www.eucast.org) for preparing the bacteria and plates. Overnight cultures were grown in BHIT at 37°C with shaking. Cultures were then diluted to a 0.5 McFarland standard (Remel, Thermo Scientific) before plating onto BHIT agar. Six different antibiotic susceptibility disks were placed on the agar (doxycycline, 30 µg; chloramphenicol, 30 µg; fosfomycin, 200 µg; clindamycin, 10 µg; vancomycin, 30 µg; penicillin G, 1 µg) (Oxoid). Plates were allowed to grow for 24 hours at 37°C before measuring zones of inhibition. Any zone of clearing around the disk was taken as susceptibility, while disks with no clearing were taken as resistant.

## ACKNOWLEDGMENTS

The authors thank Joanne Hunt for her bioinformatics support.

This work was supported by the Biotechnology and Biological Sciences Research Council grant BB/W510531/1 supporting R.H.

R.H., R.C., B.M., and G.H.T.: conceptualization; R.H., M.R., and B.M.: validation; R.H., S.M., and G.H.T.: formal analysis; R.H., S.M., and M.R.: investigation; R.H. and M.R.: data curation; H.N.W., M.R., and M.J.H.: resources; R.H.: writing–original draft; R.H., S.M., H.N.W., M.R., R.C., M.J.H., A.J.W., B.M., and G.H.T.: writing–review and editing; R.H., M.R., and S.M.: visualization; A.J.W., B.M., and G.H.T.: supervision; R.H., B.M., and G.H.T.: project administration; B.M. and G.H.T.: funding acquisition.

## AUTHOR AFFILIATIONS

[1]Department of Biology, University of York, York, United Kingdom

[2]York Biomedical Research Institute, University of York, York, United Kingdom

[3]Centre for Biomedicine, Hull York Medical School, University of Hull, Hull, United Kingdom

[4]Port Sunlight Laboratory, Unilever Research & Development, Merseyside, United Kingdom

[5]York Structural Biology Laboratory, Department of Chemistry, University of York, York, United Kingdom

## AUTHOR ORCIDs

Reyme Herman http://orcid.org/0000-0002-6620-3981
Barry Murphy http://orcid.org/0000-0003-2305-5875
Gavin H. Thomas http://orcid.org/0000-0002-9763-1313

## FUNDING

| Funder | Grant(s) | Author(s) |
|---|---|---|
| Biotechnology and Biological Sciences Research Council | BB/W510531/1 | Reyme Herman |

## AUTHOR CONTRIBUTIONS

Reyme Herman, Conceptualization, Data curation, Formal analysis, Investigation, Project administration, Validation, Visualization, Writing – original draft, Writing – review and editing | Sean Meaden, Formal analysis, Investigation, Writing – review and editing | Michelle Rudden, Data curation, Investigation, Resources, Validation, Visualization, Writing – review and editing | Robert Cornmell, Conceptualization, Writing – review and editing | Holly N. Wilkinson, Resources, Writing – review and editing | Matthew J. Hardman, Resources, Writing – review and editing | Anthony J. Wilkinson, Supervision, Writing – review and editing | Barry Murphy, Conceptualization, Funding acquisition, Project administration, Supervision, Validation, Writing – review and editing | Gavin H. Thomas, Conceptualization, Formal analysis, Funding acquisition, Project administration, Supervision, Writing – review and editing

## DATA AVAILABILITY

The 24 dereplicated genomes and associated plasmids were deposited to GenBank within BioProject PRJNA1191350.

## ETHICS APPROVAL

This study was conducted in accordance with the ethical principles of Good Clinical Practice and the Declaration of Helsinki. The local ethics committees of Unilever R&D (Port Sunlight, UK) approved the protocols before commencement of the studies and all subjects gave written informed consent.

## ADDITIONAL FILES

The following material is available online.

### Supplemental Material

**Supplemental figures and tables. (mSystems00459-25-s0001.pdf).** Fig. S1-S20; Tables S1-S7.

Open Peer Review

**PEER REVIEW HISTORY (review-history.pdf).** An accounting of the reviewer comments and feedback.

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
