## [Reviewer comments · mSystems]

Revealing the diversity of commensal corynebacteria from a single human skin site

Reyme Herman, Sean Meaden, Michelle Rudden, Robert Cornell, Holly Wilkinson, Matthew Hardman, Anthony Wilkinson, Barry Murphy, and Gavin Thomas

Corresponding Author(s): Gavin Thomas, University of York

Review Timeline:

Submission Date:	April 1, 2025
Editorial Decision:	May 23, 2025
Revision Received:	July 22, 2025
Editorial Decision:	August 18, 2025
Revision Received:	September 8, 2025
Accepted:	September 9, 2025

Editor: Fatima Pereira

Reviewer(s): Disclosure of reviewer identity is with reference to reviewer comments included in decision letter(s). The following individuals involved in review of your submission have agreed to reveal their identity: Holger Brüggemann (Reviewer #1); Irma Martinez-Flores (Reviewer #2)

Transaction Report:

DOI: <https://doi.org/10.1128/msystems.00459-25>

Re: mSystems00459-25 (**Revealing the diversity of commensal corynebacteria from a single human skin site**)

Dear Prof. Gavin H Thomas:

We have finished reviewing your manuscript. I believe the modifications requested by the reviewers are minor. The work can be accepted if all of the points raised are adequately addressed. Please see my comments below.

-A drep analysis of the 215 genomes revealed that many were redundant or nearly-identical (99.5% ANI cutoff). Please clearly indicate in the abstract how many de-replicated genomes were identified in total.

Revision Guidelines

Sincerely,
Fatima Pereira
Editor
mSystems

Reviewer #1 (Comments for the Author):

This study offers an in-depth genomic investigation into the diversity of *Corynebacterium* species at a single human skin site, the axilla. The authors expand the number of complete genomes available for this understudied genus. The identification of two potentially novel species is a notable contribution, reflecting the microbial richness that can exist even within one individual. The study includes pangenome analysis, metabolic profiling, and biosynthetic gene cluster identification. However, the work is limited by its narrow sampling-focusing on only four individuals, with the majority of isolates from one subject-which may constrain broader generalizability. Additionally, the narrow cultivation approach might exclude certain corynebacteria, potentially underestimating total diversity.

There are a few minor issues that could be clarified to improve the work.

Results

126: only 4 individuals were looked at. Limitation section

More information in the relevant MM part is needed here, regarding the bacterial cultivation: how *Corynebacterium*-specific were the used plates? How many non-corynebacteria were obtained? Fosfomycin (FF) was used. It has broad antibacterial activity against both Gram-positive and Gram-negative pathogens. What about the knowledge of FF sensitive corynebacteria?

133-144:

-So of the 150 isolates sequenced from A1: how many were clonal (=belonging to one strain)? I would suspect a large number of clonal strains.

-which to two discrete species? (line 137). Which criteria were applied to separate species (ANI<95%?) and strains (ANI of 95-99.X%) ? How was a clone defined (number of SNPs)

143/144:

One species, one strain or one clone? Since you have the WGS data you can be more specific, e.g. a SNP analysis should be done e.g. with Parsnp or other tools. Or is the NanoPore sequence quality so bad?

147: drep workflow: needs to be briefly explained, since most readers (including me) will not be familiar with it

148:

More accurate: "...genetically similar isolates, with a threshold of < 95% ANI used to distinguish between different species..." "ANI of 99.5% to capture strain diversity" I don't understand, please be more specific/accurate. If strains of the same species are concerned it can be anything between ANI 95-99.9%. It depends where you make the clonal cutoff (e.g., <50 SNPs). Again, I think you need to do a SNP analysis.

150-152: I don't know why only 16S rRNA seqs were used. You have the whole genome, so you can Blastn the whole genome to identify the species (using the DB GenBank or similar).

147-159: I think this should be done differently. SNP analysis and species identification/separation via the whole genome blast. Several tools available, e.g. here <https://www.genomicepidemiology.org/services/>

What is needed in the results: a table with the identified species in each individuals, and the number of strains and clones in each individual.

162: GTDB-tk. Please explain why this was used. GTDB-tk seems to be made for metagenome analysis. The authors have draft/complete genomes and could use tools made for WGS analyses/ comparisons. (see above).

161-175: pretty confusing to me. I would do this differently (see comment above). I think this could be done/be described more straight-forward. (to answer: of the 215 sequenced genomes: you have 7 (?) distinct species, and how many strains (per species) and clones (per strain)? (I can see from NCBI that you submitted 2 amycolatum, 1 gottgingense, 5 kefirresidentii, 10 marquesiae, 9 tuberculostearicum and 3 sp genomes. This info (if it is correct) should be clearly present in the MS and possibly also in the abstract.)

343/344: CRISPR/Cas systems were found in a few genomes. Do these genomes contain any prophage(s)?

Discussion: The discussion does not really discuss the findings of this study, in particular with regard to metabolism/functional implications. In particular lines 413-433 are more an introduction/review text than a true discussion. A limitation section could be added (e.g. only 4 participants; one cultivation method only)

Abstract: please delete "dramatically"... this isn't dramatic.

Reviewer #2 (Comments for the Author):

Minor details that may need clarification or modification:

Although I believe appropriate statistical tests have been applied, the document does not go into detail about the specific tests used and why they were chosen. It would be helpful to provide more information about the statistical methods employed, giving a justification for the statistical methods.

Deeper analysis of the functional traits of new species: While the study identifies two new species, a deeper analysis of their unique characteristics and their potential role in the skin microbiome could be explored.

Experimental validation of bioinformatics predictions: Some of the bioinformatics predictions, such as antimicrobial resistance, could benefit from experimental validation to confirm the in silico findings.

Reviewer #3 (Comments for the Author):

Herman et al. isolated and sequenced 215 closed genomes of *Corynebacterium* from the axillae of four healthy volunteers using long-read sequencing. Through comprehensive genomic analysis, they identified seven distinct species, including two novel species. By examining the gene content, they characterized and compared metabolic potential, biosynthetic gene clusters (BGCs), antimicrobial resistance genes, prophages, and phage defense systems. This is a well-executed study featuring thorough bioinformatic analysis that provides novel insights into the diversity and functional potential of *Corynebacterium* on human skin. I have only a few minor suggestions to improve clarity and readability:

Minor Comments:

1. In Figure 4, numerical values appear after the BGC names. Please clarify the meaning of these numbers in the figure legend.
2. The manuscript refers to the "abundance" of BGCs and phage defense systems. Since "abundance" typically refers to quantitative levels within samples, would "prevalence" (i.e., presence across isolate genomes) be more appropriate?
3. For the heatmap, please include a color legend indicating the range of values represented by purple and yellow.
4. Some of the bioinformatic tools used (e.g., Roary, Mash) are missing version numbers. Including these would improve reproducibility.

Dr. Fatima Pereira,
Editor,
mSystems

17th July 2025

Dear Dr. Pereira,

Thank you for handling our submission and for your nice comments about our work. We have addressed your comment and those of the two reviewers in the following text, and hope very much that with these changes the manuscript will be acceptable for publication. The editor/reviewers text is in black and our responses are in red text.

Editor's comments:

-A drep analysis of the 215 genomes revealed that many were redundant or nearly-identical (99.5% ANI cutoff). Please clearly indicate in the abstract how many de-replicated genomes were identified in total.

We are added a specific sentence to the abstract that mentions the number of dereplicated genomes identified (which was 30). This is in lines 28-31.

Reviewer #1 (Comments for the Author):

This study offers an in-depth genomic investigation into the diversity of *Corynebacterium* species at a single human skin site, the axilla. The authors expand the number of complete genomes available for this understudied genus. The identification of two potentially novel species is a notable contribution, reflecting the microbial richness that can exist even within one individual. The study includes pangenome analysis, metabolic profiling, and biosynthetic gene cluster identification. However, the work is limited by its narrow sampling-focusing on only four individuals, with the majority of isolates from one subject-which may constrain broader generalizability. Additionally, the narrow cultivation approach might exclude certain corynebacteria, potentially underestimating total diversity.

There are a few minor issues that could be clarified to improve the work.

Results

126: only 4 individuals were looked at. Limitation section

More information in the relevant MM part is needed here, regarding the bacterial cultivation: how *Corynebacterium*-specific were the used plates? How many non-corynebacteria were obtained? Fosfomycin (FF) was used. It has broad antibacterial activity

against both Gram-positive and Gram-negative pathogens. What about the knowledge of FF sensitive corynebacteria?

The selective media were not *Corynebacterium*-specific and hence the term “enrichment of *Corynebacterium*” (or variations) was used. As such, there were other non-corynebacteria that were isolated in our approach. The reviewer was right to point this out. Hence, we have added a line in the Material and Methods section indicating the other bacterial, namely two species of Staphylococci, that were isolated on the same media (line 484-5 in marked up revision).

Most *Corynebacterium* species are well known to be fosfomycin resistant (intrinsically and/or via resistance genes) (<https://doi.org/10.1128/aac.39.1.208>)

To the best of our knowledge, there are no *Corynebacterium* species that are inherently susceptible to fosfomycin. Rather, some isolates may display varying levels of resistance to fosfomycin.

133-144:

-So of the 150 isolates sequenced from A1: how many were clonal (=belonging to one strain)? I would suspect a large number of clonal strains.

Yes, there were a large number of clonal strains in the 154 isolates from A1. Using an ANI cutoff of 99.5% to define a strain, we dereplicated down to 16 strains from 154 isolates (see Table below). This threshold level was selected to be consistent with Ahmed et al (10.1128/msystems.00632-23). We have been careful not to make any assertions about how this reflects the relative abundance of these bacteria in the underarm of A1 due to the culturing step and human selection of individual colonies for sequencing.

Species	Drep secondary cluster ($\geq 99.5\%$)	Clones
C. axilliensis	1_2	6
C. kefirresidentii	2_3	1
C. kefirresidentii	2_4	2
C. kefirresidentii	2_5	59
C. tuberculostearicum	3_4	5
C. tuberculostearicum	3_5	2
C. tuberculostearicum	3_6	1
C. tuberculostearicum	3_7	1
C. tuberculostearicum	3_8	1
C. marquesiae	4_1	1
C. marquesiae	4_3	2
C. marquesiae	4_5	63
C. gottingense	5_1	6
C. amycolatum	6_1	2
C. amycolatum	6_2	1
C. jamesii	7_0	1

We acknowledge that this approach is likely to result in a lot of clonal, and hence potentially redundant isolates. This is particularly evident with *C. kefirresidentii* and *C. marquesiae*. However, *C. tuberculostearicum* isolates (same species complex) seem to be more genetically diverse. Our genetic approach remains the most comprehensive way to genetically characterise the culturable microbiome without introducing any further bias after the selective media.

-which to two discrete species? (line 137). Which criteria were applied to separate species (ANI<95%) and strains (ANI of 95-99.X%) ? How was a clone defined (number of SNPs)

The criteria used to separate species using drep is (ANI<95%) and strains (ANI of 95-99.5%). Consistent with Ahmed et al (as above), we did not use SNPs to define species. In figure 1, to which this text is linked, we have not yet described the analysis of what these species are, which is done later using GTDB-tk. There to make things more consistent with the surrounding text, the word "species" has been changed to "groups" (line 142).

143/144:

One species, one strain or one clone? Since you have the WGS data you can be more specific, e.g. a SNP analysis should be done e.g. with Parsnp or other tools. Or is the NanoPore sequence quality so bad?

Consistent with Ahmed et. al. (2023) and Jensen et al. (2023), we have chosen the genome-wide ANI approach on pyani to distinguish between different species rather than SNP analysis. It is not because SNP analysis was not possible but rather, we chose the approach that was consistent with the two recent studies on the diversity of *Corynebacterium*. As above, we place the species cutoff at (ANI = 95%). Hence, we have changed the text to "...one species dominates...". (line 150)

147: drep workflow: needs to be briefly explained, since most readers (including me) will not be familiar with it

The workflow is now briefly introduced in the next sentence (line 153 to 155)

148:

More accurate: "...genetically similar isolates, with a threshold of < 95% ANI used to distinguish between different species..."

"ANI of 99.5% to capture strain diversity" I don't understand, please be more specific/accurate. If strains of the same species are concerned it can be anything between ANI 95-99.9%. It depends where you make the clonal cutoff (e.g., <50 SNPs). Again, I think you need to do a SNP analysis.

This have now been clarified as mentioned above (line 153 to 155)

150-152: I don't know why only 16S rRNA seqs were used. You have the whole genome, so you can Blastn the whole genome to identify the species (using the DB GenBank or similar).

The 16S rRNA approach was done to show its limitations as it does not identify as many species as the WGS-based analysis. We understand the confusion with the original text. We have now changed the text between lines 156 to 162.

147-159: I think this should be done differently. SNP analysis and species identification/separation via the whole genome blast. Several tools available, e.g. here <https://www.genomicepidemiology.org/services/>
What is needed in the results: a table with the identified species in each individuals, and the number of strains and clones in each individual.

This is linked to the previous comment. The text was corrected to indicate the point of using the 16S rRNA approach (i.e. that is what most studies tend to do and we are highlighting its major limitations). We agree that a table to summarise the identified species and subsequent number of strains from each individual would be useful to have. Hence, we have added four Tables in the Supplementary Material (Supplementary Tables 1-4), mentioned in a new sentence in the text (214 and 216).

162: GTDB-tk. Please explain why this was used. GTDB-tk seems to be made for metagenome analysis. The authors have draft/complete genomes and could use tools made for WGS analyses/ comparisons. (see above).

While it is true that GTDB-tk was suggested to be a tool to aid in the analysis of metagenomic datasets, it functions as a powerful tool to assign species to assembled genomes, be it culture or metagenomics derived. This program's well-maintained database of reference genomes and ANI comparison approach provides a platform to accurately speciate any new bacterial genome and help identify truly novel species as observed in our study with *C. axilliensis* and *C. jamesii*. For example, a similar approach was performed by Salamzade *et al.* (<https://doi.org/10.1128/spectrum.03578-22>) to speciate other isolates of the genus *Corynebacterium*.

For a SNP analysis approach to be comparable, it requires a large enough database encompassing all known species of this genus. We do not believe this is feasible or is it worth the time and effort as a powerful tool like GTDB-tk already exists for the same purpose.

161-175: pretty confusing to me. I would do this differently (see comment above). I think this could be done/be described more straight-forward. (to answer: of the 215 sequenced genomes: you have 7 (?) distinct species, and how many strains (per species) and clones (per strain)? (I can see from NCBI that you submitted 2 amycolatum, 1 gottingense, 5 kefirresidentii, 10 marquesiae, 9 tuberculostearicum and 3 sp genomes. This info (if it is correct) should be clearly present in the MS and possibly also in the abstract.)

As sensibly suggested by the reviewer we have now added all these details in the four Supplementary Tables mentioned previously (Supp Table 1-4), so a reader can clearly see the relationships between the 215 sequenced genomes and what strain they were binned into. The purpose of this paragraph in the text is to consider the outputs of GTDB-tk. There were some ambiguities (e.g. *C. aurimucosum*_E becoming *C. marquesiae*) that needed to be clarified.

343/344: CRISPR/Cas systems were found in a few genomes. Do these genomes contain any prophage(s)?

The original manuscript contains a significant section on the prophage content of the new genomes, which was perhaps missed by the reviewer? These are lines 329-349 in the marked-up revision.

Discussion: The discussion does not really discuss the findings of this study, in particular with regard to metabolism/functional implications. In particular lines 413-433 are more an introduction/review text than a true discussion. A limitation section could be added (e.g. only 4 participants; one cultivation method only)

We agree the discussion lacked discussion of the metabolic implications and we have added some material on this (lines 441-447 on the markup version). We also found an incomplete sentence at the end of this text. This has been rectified. (Line 442 to 444).

We agree that a limitation section should be included in the paper and have added additional text to reflect this in the discussion (lines in 441-447 in marked up revision).

Abstract: please delete "dramatically"... this isn't dramatic.

We have removed the work "dramatically" as suggested.

Reviewer #2 (Comments for the Author):

Minor details that may need clarification or modification:

Although I believe appropriate statistical tests have been applied, the document does not go into detail about the specific tests used and why they were chosen. It would be helpful to provide more information about the statistical methods employed, giving a justification for the statistical methods.

In a highly descriptive paper like this one, we are not clear to which use of statistics the reviewer refers? The only part of our analysis where this could be relevant, we think, is the statistical support for the phylogenetic trees. For these we used approximate likelihood ratios test for tree support, rather than bootstrapping, as this is a recognised (see Anisimova and Gascuel (2006) (DOI: 10.1080/10635150600755453) method that is more suited to large datasets like those in this study. We have added the reference to the methods about this (line 530 in marked up revision).

Deeper analysis of the functional traits of new species: While the study identifies two new

species, a deeper analysis of their unique characteristics and their potential role in the skin microbiome could be explored.

As part of the analysis of all the genomes from these underarm communities we do compare predicted metabolic traits and mention some features of our two putative new species (line 221-228 in the marked up revision). The focus of this paper was to describe the corynebacterial composition of single individuals with WGS and strain isolation, providing this information openly for the community. We hope others might build on our proposed new species and undertake additional work to define them more formally through a battery of phenotypic assays. Genomes are available at the York Skin Microbiome site within MORF (<https://morf-db.org/projects/Public/MORF000083>) and strains can be requested from this collection.

Experimental validation of bioinformatics predictions: Some of the bioinformatics predictions, such as antimicrobial resistance, could benefit from experimental validation to confirm the in silico findings.

We agree that this is the most likely next project to continue this work, to work towards a functional understanding of community interactions, both beneficial and antagonistic. However, for this genomics-based study we felt that this is beyond the scope of this work.

Reviewer #3 (Comments for the Author):

Herman et al. isolated and sequenced 215 closed genomes of *Corynebacterium* from the axillae of four healthy volunteers using long-read sequencing. Through comprehensive genomic analysis, they identified seven distinct species, including two novel species. By examining the gene content, they characterized and compared metabolic potential, biosynthetic gene clusters (BGCs), antimicrobial resistance genes, prophages, and phage defense systems. This is a well-executed study featuring thorough bioinformatic analysis that provides novel insights into the diversity and functional potential of *Corynebacterium* on human skin. **I have only a few minor suggestions** to improve clarity and readability:

Minor Comments:

1. In Figure 4, numerical values appear after the BGC names. Please clarify the meaning of these numbers in the figure legend.

We realise that this was not very clear and have changed this so it should be both clearer and explained in full in the legend.

2. The manuscript refers to the "abundance" of BGCs and phage defense systems. Since "abundance" typically refers to quantitative levels within samples, would "prevalence" (i.e., presence across isolate genomes) be more appropriate?

We agree and have made the changes where appropriate.

3. For the heatmap, please include a color legend indicating the range of values represented by purple and yellow.

This has now been added to Fig 3 and Supplementary Fig 10.

4. Some of the bioinformatic tools used (e.g., Roary, Mash) are missing version numbers. Including these would improve reproducibility.

Version numbers for bioinformatics tools were added to the Materials and Method when missing.

Yours sincerely,

Gavin H. Thomas & Reyme Herman (On behalf of all authors)

Re: mSystems00459-25R1 (**Revealing the diversity of commensal corynebacteria from a single human skin site**)

Dear Prof. Gavin H Thomas:

The reviewer's feedback was that the comments made have been largely addressed. However, Reviewer #2 points out that the ecological implications of the genetic and functional differences between the identified *Corynebacterium* species need to be discussed in more detail. Please include this in the discussion. When responding to the reviewers' comments, please provide a detailed description of the amendments made to the manuscript in response to Reviewer #2, as well as to the points I raise below.

-Abstract: "Isolate dereplication then revealed 30 genetically distinct strains spanning seven distinct species, including two novel ones provisionally named *C. axilliensis* and *C. jamesii*"

It would be more accurate to say that 'genomes' rather than 'isolates' were dereplicated. I suggest rephrasing as follows: "The study yielded 215 closed genomes, comprising 30 distinct representative genomes following dereplication. These genomes span seven distinct species, including two new species that have been given the provisional names *C. axilliensis* and *C. jamesii*."

-Line 138: "isolates" should not be capitalised.

-Lines 151-152: Please provide a reference for the definition of strain presented (95%-99.5% ANI), and clearly justify your choice in the text.

-Lines 164-166: Are the colours referring to supplementary Fig. 3? I suggest describing colour codes in the figure legend instead of the main text.

-Please include a brief explanation in materials and methods for the provisional names chosen for the new species, i.e. "axilliensis" and "jamesii".

-Lines 549-550, EUCAST test: please provide a reference (or link). Please also indicate how quality control was implemented in materials and methods.

-Line 454: "revealed extensive genomic variation across all 215 closed genomes" : Figure 1 and ANI analyses show that a large proportion of the genomes are nearly 100% identical, so this claim sounds like an overstatement, please rephrase.

Figure 5: In the heatmap axis, please replace "system" by "Phage defence system", for easier interpretation by the reader. The legend indicates that colours refer to prevalence - how was this defined? In addition, the figure shows "system count"- not prevalence. Please explain this discrepancy.

Table 1: The legend refers to colours of prophage names, but these are lacking on the table; please correct.

Revision Guidelines

Sincerely,
Fatima Pereira
Editor
mSystems

Reviewer #1 (Comments for the Author):

the authors have addressed the issues raised by the reviewer(s)

Reviewer #2 (Comments for the Author):

General Comments: The study presents a comprehensive analysis of *Corynebacterium* diversity in the human axilla using whole-genome sequencing (WGS) and culture-based methods. The discovery of novel species and the detailed characterization of genetic and functional diversity are valuable contributions to the field of the cutaneous microbiome.

The work concisely reflects the impact and novelty of the study, such as the discovery of new species and their potential functional differentiation.

A comprehensive description summarizing the observations that differentiate the species could be added. Also, briefly justify the study, highlighting the importance of understanding diversity for skin health and the improvement compared to previous research on *Corynebacterium*, which could also be added.

The key points this study brings to the table, making it recommendable work, are:

Discovery of Novel Species: Identification of two potentially novel *Corynebacterium* species (*C. axilliensis* and *C. jamesii*) highlights that our knowledge of the cutaneous microbiome is still incomplete.

Deep dive into the genetic and functional differences among *Corynebacterium* isolates, including metabolic pathways, antimicrobial resistance genes, and phage defense systems.

Culture-Based WGS Approach provides a cost-effective way to bypass the limitations of 16S rRNA sequencing and capture a broader spectrum of genetic information.

The generated collection of complete *Corynebacterium* genomes serves as a valuable resource for future studies investigating the diversity, function, and ecological roles of this important skin genus.

Reviewer #3 (Comments for the Author):

All of my previous comments have been satisfactorily addressed in this resubmission. I have no further concerns.

Dr. Fatima Pereira,
Editor,
mSystems

4th September 2025

Re: mSystems00459-25R1 (Revealing the diversity of commensal corynebacteria from a single human skin site)

Dear Dr. Pereira,
Thank you for handling the revision of our submission. We are happy that both **reviewer 1 and reviewer 3 were happy with our changes** and have no further comments.

In the following pages we address **your comments and those further questions from Reviewer 2.**

We hope that with these changes the manuscript will be acceptable for publication. The editor/reviewers text is in black and our responses are in blue text.

Yours sincerely,

Gavin H. Thomas & Reyme Herman (On behalf of all authors)

Comments from the Editor

The reviewer's feedback was that the comments made have been largely addressed. However, Reviewer #2 points out that the **ecological implications of the genetic and functional differences between the identified *Corynebacterium* species need to be discussed in more detail. Please include this in the discussion.** When responding to the reviewers' comments, please provide a detailed description of the amendments made to the manuscript in response to Reviewer #2, as well as to the points I raise below.

-Abstract: "Isolate dereplication then revealed 30 genetically distinct strains spanning seven distinct species, including two novel ones provisionally named *C. axilliensis* and *C. jamesii*"

It would be more accurate to say that 'genomes' rather than 'isolates' were dereplicated. I suggest rephrasing as follows: "The study yielded 215 closed genomes, comprising 30 distinct representative genomes following dereplication. These genomes span seven distinct species, including two new species that have been given the provisional names *C. axilliensis* and *C. jamesii*."

Thanks you for this suggestion, which we have changed (lines 27-29). We have also included the point about species not previously linked to the skin (viz. *C. gottingense*).

-Line 138: "isolates" should not be capitalised.

Corrected.

-Lines 151-152: Please provide a reference for the definition of strain presented (95%-99.5% ANI), and clearly justify your choice in the text.

We have added a line with references to support this (lines 151-152).

-Lines 164-166: Are the colours referring to supplementary Fig. 3? I suggest describing colour codes in the figure legend instead of the main text.

The colours have been removed from the main text (line 170). The figure legend for Supplementary Fig 3 has also been updated.

-Please include a brief explanation in materials and methods for the provisional names chosen for the new species, i.e, "axilliensis" and "jamesii".

This has been added (lines 540-542).

-Lines 549-550, EUCAST test: please provide a reference (or link). Please also indicate how quality control was implemented in materials and methods.

The link to EUCAST has now provided (line 557) and additional details to the Table and legend about the concentrations of antibiotic used in each case. We only used the EUCAST protocols for growing the bacteria and preparing the plates as we did

not have the standard strains for calibration of the clinical test. We have more clearly defined what has been measured (no growth inhibition or growth inhibition around the disk) and have changed the text to acknowledge that this only allows us to compare data between strains and not absolute quantitation of resistance.

-Line 454: "revealed extensive genomic variation across all 215 closed genomes" : Figure 1 and ANI analyses show that a large proportion of the genomes are nearly 100% identical, so this claim sounds like an overstatement, please rephrase.

We have altered this sentence to more reflect what we observed (lines 459-460)

Figure 5: In the heatmap axis, please replace "system" by "Phage defence system", for easier interpretation by the reader. The legend indicates that colours refer to prevalence - how was this defined? In addition, the figure shows "system count"- not prevalence. Please explain this discrepancy.

We have made these suggested changes to Fig. 5. The editor is correct that "System count" is the wrong terminology, which we have changed to "Prevalence".

Table 1: The legend refers to colours of prophage names, but these are lacking on the table; please correct.

This table was initially prepared with colours, but have been removed, hence the mentions of colours have been removed from the text.

Comments from Reviewer #2 (Comments for the Author):

General Comments: The study presents a comprehensive analysis of *Corynebacterium* diversity in the human axilla using whole-genome sequencing (WGS) and culture-based methods. The discovery of novel species and the detailed characterization of genetic and functional diversity are valuable contributions to the field of the cutaneous microbiome.

The work concisely reflects the impact and novelty of the study, such as the discovery of new species and their potential functional differentiation.

We thank the reviewer(s) for taking the time to review the responses.

A comprehensive description summarizing the observations that differentiate the species could be added.

We now recapped the major differences between the species in the discussion, in particular the paragraph starting at line 427.

Also, briefly justify the study, highlighting the importance of understanding diversity for skin health and the improvement compared to previous research on *Corynebacterium*, which could also be added.

The justification for the study was mentioned in the last paragraph of the introduction, which was to look at a particular site in real depth, which in this case

was the axilla, based on our long-term interests in the microbiology of body odour production. The focus on the axilla itself is different to other studies, and combined with our culture and sequence approach this study has significant improvement over other studied completed previously (mentioned in the last paragraph in the discussion).

The key points this study brings to the table, making it recommendable work, are:

Discovery of Novel Species: Identification of two potentially novel *Corynebacterium* species (*C. axilliensis* and *C. jamesii*) highlights that our knowledge of the cutaneous microbiome is still incomplete.

Deep dive into the genetic and functional differences among *Corynebacterium* isolates, including metabolic pathways, antimicrobial resistance genes, and phage defense systems.

Culture-Based WGS Approach provides a cost-effective way to bypass the limitations of 16S rRNA sequencing and capture a broader spectrum of genetic information.

The generated collection of complete *Corynebacterium* genomes serves as a valuable resource for future studies investigating the diversity, function, and ecological roles of this important skin genus.

Re: mSystems00459-25R2 (**Revealing the diversity of commensal corynebacteria from a single human skin site**)

Dear Prof. Gavin H Thomas:

Your manuscript has been accepted, and I am forwarding it to the ASM production staff for publication. Your paper will first be checked to make sure all elements meet the technical requirements. ASM staff will contact you if anything needs to be revised before copyediting and production can begin. Otherwise, you will be notified when your proofs are ready to be viewed.

Sincerely,
Fatima Pereira
Editor
mSystems